# System Combination via Quality Estimation for Grammatical Error Correction

**Muhammad Reza Qorib** and **Hwee Tou Ng**
Department of Computer Science, National University of Singapore
mrqorib@u.nus.edu, nght@comp.nus.edu.sg

## Abstract

Quality estimation models have been developed to assess the corrections made by grammatical error correction (GEC) models when the reference or gold-standard corrections are not available. An ideal quality estimator can be utilized to combine the outputs of multiple GEC systems by choosing the best subset of edits from the union of all edits proposed by the GEC base systems. However, we found that existing GEC quality estimation models are not good enough in differentiating good corrections from bad ones, resulting in a low $F_{0.5}$ score when used for system combination. In this paper, we propose GRECO[1], a new state-of-the-art quality estimation model that gives a better estimate of the quality of a corrected sentence, as indicated by having a higher correlation to the $F_{0.5}$ score of a corrected sentence. It results in a combined GEC system with a higher $F_{0.5}$ score. We also propose three methods for utilizing GEC quality estimation models for system combination with varying generality: model-agnostic, model-agnostic with voting bias, and model-dependent method. The combined GEC system outperforms the state of the art on the CoNLL-2014 test set and the BEA-2019 test set, achieving the highest $F_{0.5}$ scores published to date.

## 1 Introduction

Grammatical error correction (GEC) is the task of automatically detecting and correcting errors in text, including but not limited to grammatical errors, misspellings, orthographic errors, and semantic errors (Chollampatt et al., 2016; Chollampatt and Ng, 2018a; Qorib et al., 2022; Bryant et al., 2023). A GEC model is evaluated by calculating the $F$-score (van Rijsbergen, 1979) from comparing the edits proposed by the GEC model against gold (human-annotated) reference edits. GEC edits are a set of insertion, deletion, or substitution operations that are applied to the original (source) sentence to make it free from errors. An edit is represented by three values: start index, end index, and correction string (Table 1). Since CoNLL-2014 (Ng et al., 2014), $F_{0.5}$ has become the standard metric for GEC. Grundkiewicz et al. (2015) and Chollampatt and Ng (2018c) reported that the $F_{0.5}$ score correlates better with human judgment than other GEC metrics.

Qorib and Ng (2022) reported that GEC models have outperformed humans when measured by the $F_{0.5}$ metric, but still make occasional mistakes in simple cases. Thus, we need a way to evaluate the corrections proposed by a GEC model before accepting their corrections as a replacement for our original sentences. In real-world use cases where the gold reference is not available, we can use a GEC quality estimation model to assess the quality of the correction made by a GEC model.

GEC quality estimation model accepts a source sentence and its correction and produces a quality score. The quality score characterizes the accuracy and appropriateness of a correction with respect to the source sentence. The higher the score, the more accurate and appropriate the correction is. A quality estimation model is typically used as a filtering method to accept or reject a correction made by a GEC model (Chollampatt and Ng, 2018b). It can also be used for choosing the best correction from the top-$k$ outputs of a GEC system (Liu et al., 2021).

In this paper, we propose to extend that use case further. Instead of choosing the best correction (hypothesis) from GEC models, we can use a GEC quality estimation model to produce a new and more accurate correction, based on the edits that appear in the hypotheses. We generate all possible hypotheses from the edit combinations and score them using a quality estimation model. The highest-scoring hypothesis is then deemed as the most appropriate correction of the source sentence.

---

[1]Source code available at https://github.com/nusnlp/greco.

| Source | To sum it up I still consider having their own car is way more safe and convinient . |
|---|---|
| Correction | To sum up , I still consider having your own car way more safe and convenient . |
| Differences | To sum {it} up {,} I still consider having {their→ **your**} own car {is} way more safe and {convinient → **convenient**} . |
| Edits | (2, 3, ''),   (4, 4, ','),   (8, 9, 'your'),   (11, 12, ''),   (16, 17, 'convenient') |

Table 1: Example GEC edits.

We discuss this in more detail in Section 3.

The main contributions of this paper are:

- We present novel methods for utilizing GEC quality estimation models for system combination.

- We reveal and highlight the low performance of existing GEC quality estimation models when used for system combination.

- We present a new state-of-the-art GEC quality estimation model that has better correlation to the $F_{0.5}$ score and produces higher $F_{0.5}$ scores when used for system combination.

- We report new state-of-the-art scores on the CoNLL-2014 and BEA-2019 test sets.

## 2 Related Work

### 2.1 GEC Quality Estimation Models

In this section, we briefly discuss existing neural GEC quality estimation models, including a neural reference-less GEC metric.

#### 2.1.1 NeuQE

NeuQE (Chollampatt and Ng, 2018b) is the first neural quality estimation model for GEC. NeuQE uses the predictor-estimator framework (Kim et al., 2017) which trains a word prediction task on the predictor network and trains the quality score on the estimator network. The estimator is trained using knowledge from the predictor. NeuQE has two types of model, one for $F_{0.5}$ score estimation and the other for post-editing effort estimation. NeuQE is trained on the NUCLE (Dahlmeier et al., 2013) and FCE (Yannakoudakis et al., 2011) corpora.

#### 2.1.2 VERNet

VERNet (Liu et al., 2021) estimates the quality of a GEC model from the top-$k$ outputs of beam search decoding of the GEC model. VERNet uses BERT-like architecture to get the representation of each token. It then constructs a fully-connected graph between pairs of (source, hypothesis) for each beam search output to learn the interaction between hypotheses, then summarizes and aggregates the information of the hypotheses' interaction using two custom attention mechanisms. VERNet trains the model using the top-5 outputs of the Riken&Tohoku (Kiyono et al., 2019) model on the FCE, NUCLE, and W&I+LOCNESS (Bryant et al., 2019; Granger, 1998) datasets.

#### 2.1.3 SOME

SOME (Yoshimura et al., 2020) is a reference-less GEC metric that scores a GEC correction based on three scoring aspects: grammaticality, fluency, and meaning preservation. SOME consists of three BERT models, one for each scoring aspect. Different from the aforementioned GEC quality estimation models, SOME does not aim to estimate the $F_{0.5}$ score. Instead, it estimates the aspect scores directly. The authors created a new dataset to train the BERT models by annotating outputs of various GEC systems on the CoNLL-2013 test set with the three scoring aspects. The authors argue that reference-less metrics are better than $F_{0.5}$ score because it is difficult to cover all possible corrections in the gold reference.

### 2.2 GEC System Combination Methods

In this section, we briefly discuss state-of-the-art GEC system combination methods.

#### 2.2.1 ESC

ESC (Qorib et al., 2022) is a system combination method that takes the union of all edits from the base systems, scores each edit to decide whether the edit should be kept or discarded, and generates the final corrections using the selected edits. ESC uses logistic regression to score each edit based on the edit type and inclusion in the base systems, and filters the overlapping edit based on a threshold and a greedy selection method. ESC is trained on the BEA-2019 development set.

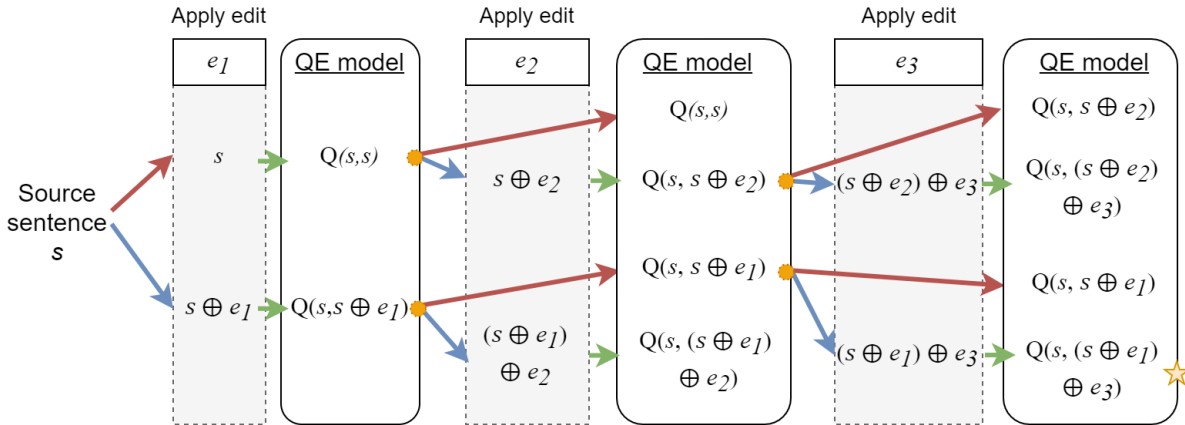

Figure 1: Beam search with beam size $(b) = 2$. The blue arrow denotes generation of a new hypothesis, and the orange circle denotes the hypotheses with the highest scores. At each step, new hypotheses are generated by applying edit $e_i$ to the top-$b$ hypotheses of step $i - 1$.

### 2.2.2 MEMT

MEMT (Heafield and Lavie, 2010) is a system combination method that combines models' outputs by generating candidate hypotheses through token alignments and scoring each candidate according to its textual features, which include n-gram language model score, n-gram similarity to each base model's output, and sentence length. MEMT was originally designed for machine translation system combination, but Susanto et al. (2014) successfully adapted it for use in GEC.

### 2.2.3 EditScorer

EditScorer (Sorokin, 2022) is a model that scores each edit based on its textual features to generate a better correction. The model can be used to re-rank edits from a single model or combine edits from multiple models. It has a similar principle to ESC but uses the textual features of the edit and its surrounding context instead of the edit type. The textual feature is acquired from RoBERTa-large's (Liu et al., 2019) token representation of the candidate sentence. The model is trained with more than 2.4M sentence pairs from cLang8 (Rothe et al., 2021) and the BEA-2019 training set.

## 3 GEC System Combination via Quality Estimation

To be able to use a GEC quality estimation model for system combination, we assume an ideal quality estimation model that can discern good hypotheses from bad ones and produce appropriate quality scores. Even though a perfect quality estimation model does not exist yet, quality estimation that

behaves closely to our assumption will be good enough to be useful for combining GEC systems.

### 3.1 Problem Formulation

For a source sentence $s = \{s_1, s_2, ..., s_l\}$ with length $l$ and a hypothesis $h = \{h_1, h_2, ..., h_m\}$ with length $m$, a quality estimation model produces a quality score $Q(s, h)$ to assess how good $h$ is as a correction to $s$. When combining GEC systems, we have multiple hypotheses from different base GEC systems. From these hypotheses, we can extract all the edits. Let $\mathbb{E}$ denote the union of all edits.

A new hypothesis can be generated by applying an edit $e_i \in \mathbb{E}$ to the source sentence $s$. If it is a correct edit ($e_i^+$), the quality score of the resulting hypothesis should be higher than when the edit is not applied or when a wrong edit ($e_i^-$) is applied. Let $h \oplus e$ denote the operation of applying edit $e$ to sentence $h$. For any hypothesis $h$ (including the case of $h = s$), an ideal quality estimation model should have the following property:

$$Q(s, h \oplus e^+) > Q(s, h) > Q(s, h \oplus e^-) \quad (1)$$

### 3.2 Beam Search

From an edit union of size $|\mathbb{E}|$, we can get $2^{|\mathbb{E}|}$ possible hypotheses. However, scoring all possible hypotheses is too costly, so we use beam search with size $b$ to generate the potential candidates in a reasonable time. We apply each edit in $\mathbb{E}$ one by one to the hypotheses in the current beam to generate new candidates, with time complexity $O(b \times |\mathbb{E}|)$.

Initially, the beam contains the source sentence and all edits in $\mathbb{E}$ are sorted from left to right, i.e.,

edits with smaller start and end indices are processed earlier. In each step, we generate new candidates by applying the current edit to all candidate sentences in the beam if it does not create a conflict with previously added edits. We use the edit conflict definition of Qorib et al. (2022). Next, we compute the quality scores for the new candidates and add them to the beam. At the end of each step, we trim the beam by only keeping the top-$b$ candidates with the highest quality scores. After we finish processing all edits, the candidate with the highest quality score becomes the final correction. We illustrate this process in Figure 1.

## 4 Quality Estimation Method

A correction produced by a GEC model can be wrong in three aspects: keeping wrong words or phrases from the source sentence, changing the words or phrases into the wrong ones, or missing some words or phrases. In other words, a quality estimation model needs to know which words are correct and which are wrong, as well as determine whether the gaps between words are correct or wrong (in which case a word or phrase needs to be inserted).

A GEC quality estimation model should also produce the quality scores proportionately. A better correction of the same source sentence should get a higher score than a worse one. That is, a quality estimation model should be able to rank hypotheses by their quality scores correctly.

In this section, we describe our approach to build a quality estimation model with the aforementioned qualities, which we call **GRECO** (**G**rammaticality-scorer for **re**-ranking **co**rrections).

### 4.1 Architecture

Our model uses a BERT-like pre-trained language model as its core architecture (Figure 2). We draw inspiration from quality estimation for machine translation models (Kim et al., 2019; Lee, 2020; Wang et al., 2020). The input to the model is the concatenation of the source sentence and a hypothesis, with the source sentence and the hypothesis prefixed with [CLS] ($s_0$) and [SEP] ($h_0$) pseudo-tokens respectively.

For every word in the hypothesis, the model learns to predict its word label $w_i$ and gap label $g_i$. The word label denotes whether the current word is correct. $w_i$ is 1 when the word is correct and 0 otherwise. The gap label denotes whether there

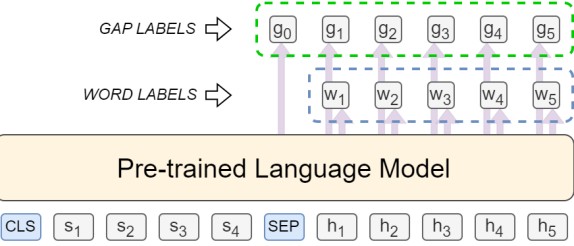

Figure 2: Model architecture

should be a word or phrase inserted right after the current word and before the next word. $g_i$ is 1 when the gap is correct (i.e., there should be no words inserted) and 0 otherwise. The gold word labels and gap labels are computed by extracting the differences between the hypothesis and the gold reference sentence using ERRANT (Bryant et al., 2017). The word label $w_i$ and gap label $g_i$ are computed from the projection of the embeddings learned by the pre-trained language model to a value in [0,1] using a two-layered neural network with tanh activation ($\phi$). We formally describe them in Equation (2) and (3), where LM denotes the pre-trained language model, $\sigma$ denotes the sigmoid function, $\boldsymbol{A}_w$, $\boldsymbol{a}_w$, $\boldsymbol{b}_w$, and $b_w$ denote the weights and biases for the weight label projector, and $\boldsymbol{A}_g$, $\boldsymbol{a}_g$, $\boldsymbol{b}_g$, and $b_g$ denote the weights and biases for the gap label projector. The size of $\boldsymbol{A}_w$ and $\boldsymbol{A}_g$ is $d_{LM} \times d_{LM}$, while the size of $\boldsymbol{a}_w$, $\boldsymbol{a}_g$, $\boldsymbol{b}_w$, and $\boldsymbol{b}_g$ is $d_{LM} \times 1$, with $d_{LM}$ being the language model dimension.

$$
\begin{aligned}
\boldsymbol{V} &= \text{LM}(s; h) \\
&= \text{LM}(s_0, s_1, \ldots, s_l, h_0, h_1, \ldots, h_m) \\
&= \{\boldsymbol{v}_0^s, \boldsymbol{v}_1^s, \ldots, \boldsymbol{v}_l^s, \boldsymbol{v}_0^h, \boldsymbol{v}_1^h, \ldots, \boldsymbol{v}_m^h\}
\end{aligned}
$$

$$
w_i = \sigma(\boldsymbol{a}_w^T \phi(\boldsymbol{A}_w \boldsymbol{v}_i^h + \boldsymbol{b}_w) + b_w) \tag{2}
$$

$$
g_i = \sigma(\boldsymbol{a}_g^T \phi(\boldsymbol{A}_g \boldsymbol{v}_i^h + \boldsymbol{b}_g) + b_g) \tag{3}
$$

The length of the word label vector $\boldsymbol{w}$ is the same as the hypothesis ($m$), while the length of the gap label vector $\boldsymbol{g}$ is $m + 1$. The gap vector length is one more than the hypothesis' length to account for the potentially missing words at the start of the hypothesis. If the pre-trained language model uses sub-word tokenization, tokens that are not the beginning of a word are masked. The quality score $Q(s, h)$ is calculated from the normalized product of the word label and gap label probabilities from all words in the hypothesis.

$$
Q(s, h) = \sqrt[2m+1]{\prod_{i=1}^{m} w_i \cdot \prod_{i=0}^{m} g_i} \tag{4}
$$

## 4.2 Loss Function

The model is trained on two objectives: predicting the word label and gap label, and ranking the hypotheses correctly with the quality score, i.e., hypotheses with higher $F_{0.5}$ scores should have higher quality scores than hypotheses with lower $F_{0.5}$ scores. This translates into three loss functions: word label loss ($\mathcal{L}_w$), gap label loss ($\mathcal{L}_g$), and rank loss ($\mathcal{L}_r$). The first two losses are based on binary cross-entropy loss and the rank loss is based on RankNet (Burges et al., 2005; Burges, 2010) with a slight modification to amplify the power term with a multiplier $\mu$.

$$\mathcal{L} = \frac{1}{n} \sum_{j=1}^{n} \mathcal{L}_{w(j)} + \frac{1}{n} \sum_{j=1}^{n} \mathcal{L}_{g(j)} + \gamma \cdot \mathcal{L}_r \quad (5)$$

$$\mathcal{L}_w = -\frac{1}{m} \sum_{i=1}^{m} (y_i^w \cdot \log w_i + \quad (6)$$
$$(1 - y_i^w) \cdot \log(1 - w_i))$$

$$\mathcal{L}_g = -\frac{1}{m+1} \sum_{i=0}^{m} (y_i^g \cdot \log g_i + \quad (7)$$
$$(1 - y_i^g) \cdot \log(1 - g_i))$$

$$\mathcal{L}_r = \sum_{y_v^r > y_u^r} \log \left( 1 + e^{-\sigma(Q_v - Q_u) \cdot \mu} \right) \quad (8)$$

We formalize the loss functions in Equation (5) to (8), where $n$ is the number of training instances, $y^w$ and $y^g$ are the correct labels for the word label and gap label respectively, $y_v^r$ and $Q_v$ are the $F_{0.5}$ score and quality score of hypothesis $v$ respectively, and $\gamma$ is a hyper-parameter.

## 4.3 System Combination Biases

The $Q$ score from quality estimation models is model-agnostic as it fully depends on the source sentence and the hypothesis sentence, independent of the system that proposes the hypothesis. With a perfect quality estimation model, it should be enough to get the best hypothesis. If we use an imperfect quality estimation model, some valuable information in the system combination task can be useful to get a better hypothesis, such as how many systems propose an edit and which systems propose it. We incorporate the former through a voting bias and the latter through edit scores from an edit-based system combination method. In this section, we discuss how we replace the quality score $Q$ in the beam search with a biased hypothesis score $Q'$.

## 4.3.1 Voting Bias

Model voting is a common ensemble method that chooses a prediction label based on how many base systems predict that label. The rationale behind it is straightforward: the more systems propose a label, the more likely for it to be correct. In GEC, voting ensemble has also been used to combine edit labels from multiple GEC sequence-tagging models (Tarnavskyi et al., 2022).

$$h = s \oplus \mathbb{E}_h \quad (9)$$
$$Q'(s, h) = Q(s, h) \cdot V(\mathbb{E}_h)^\alpha \quad (10)$$
$$V(\mathbb{E}_h) = \frac{1}{|\mathbb{E}_h|} \sum_{e \in \mathbb{E}_h} \frac{count(e)}{c} \quad (11)$$

We incorporate voting bias into beam search by multiplying the quality score with a voting score $V$ (Eq. 10). The voting score is calculated by the average number of base systems that propose an edit ($count(e)$) for all edits in the hypothesis ($\mathbb{E}_h$), normalized by the number of base systems $c$. The effect of voting bias is governed by a hyper-parameter $\alpha$, $0 \leq \alpha \leq 1$. If $\alpha = 0$, voting bias is not used.

## 4.3.2 Edit Score

Qorib et al. (2022) reported that the best hypothesis is the one that maximizes the strengths of the base systems. Edit-based GEC system combination methods approximate the strengths of GEC models through their performance on each edit type and learn the best combination from the edit type feature. If we have edit scores that reflect the base GEC models' strength, we can incorporate them into the hypothesis scoring function.

One way to incorporate the edit scores is by multiplying the hypothesis score with the edit scores. However, if we only multiply the scores of the edits that are applied to the hypothesis, we will reward hypotheses with fewer edits, even if we normalize the edit score[2]. Instead, we want to reward hypotheses that contain good edits and penalize hypotheses that miss good edits. Thus, we design the edit score to be the product of all edits in the edit union $\mathbb{E}$.

$$Q'(s, h) = Q(s, h)^{1-\beta} \cdot V(\mathbb{E}_h)^\alpha \cdot ES(\mathbb{E}_h, \mathbb{E})^\beta \quad (12)$$

---

[2]For example, if edit $e_1$ has an edit score of 0.95 and edit $e_2$ has an edit score of 0.9, the score of applying both edits is lower than just applying $e_1$ if we only multiply the scores of edits that appear in the hypothesis, even though $e_2$ is also a good edit.

| Model | Single system evaluation | | | | Multi-system evaluation | | | |
|---|---|---|---|---|---|---|---|---|
| | $\rho$ | **P** | **R** | **F$_{0.5}$** | $\rho$ | **P** | **R** | **F$_{0.5}$** |
| NeuQE | –0.003 | 52.53 | 12.83 | 32.45 | 0.212 | 38.30 | 10.84 | 25.43 |
| VERNet | 0.199 | 72.13 | 35.93 | 60.04 | 0.354 | 65.06 | 22.41 | 47.12 |
| SOME | 0.002 | 53.02 | 51.06 | 52.62 | 0.392 | 47.30 | 34.62 | 44.07 |
| GPT-2 | 0.088 | 54.09 | 50.98 | 53.43 | 0.116 | 46.67 | 33.32 | 43.21 |
| **GRECO** | **0.445** | 71.23 | 47.72 | **64.84** | **0.415** | 67.39 | 30.71 | **54.40** |

Table 2: Quality estimation and re-ranking results on the CoNLL-2014 test set. $\rho$ denotes Spearman's rank correlation coefficient.

$$p_{edit}(e, \mathbb{E}_h) = \begin{cases} p_{ES}(e) & \text{if } e \in \mathbb{E}_h \\ 1 - p_{ES}(e) & \text{otherwise} \end{cases} \quad (13)$$

$$ES(\mathbb{E}_h, \mathbb{E}) = \sqrt[|\mathbb{E}|]{\prod_{e \in \mathbb{E}} p_{edit}(e, \mathbb{E}_h)} \quad (14)$$

We formulate the hypothesis score with voting bias and edit score $ES$ in Equation (12) – (14), where $p_{ES}$ denotes the probability of each edit. The effect of edit score is governed by the hyper-parameter $\beta$, $0 \le \beta < 1$. If $\beta = 0$, the edit score is not used.

## 5 Experiments

### 5.1 Model Training

We use DeBERTA-V3-Large (He et al., 2023) as the pre-trained language model. We train the quality estimation model using the unique corrections of nine GEC systems, which are BART-GEC (Katsumata and Komachi, 2020), GECToR RoBERTa (Omelianchuk et al., 2020), GECToR XLNet, GECToR BERT, Kakao&Brain ensemble (Choe et al., 2019), Kakao&Brain Transformer-base, Riken&Tohoku ensemble (Kiyono et al., 2019), T5-Large (Rothe et al., 2021), and UEDIN-MS ensemble (Grundkiewicz et al., 2019), on the W&I+LOCNESS training set.

The training data are grouped into small groups of size $n$, with corrections of the same source sentence grouped together as much as possible, and each group must contain at least $\frac{n}{2}$ corrections from the same source sentence. This way, the rank loss (Eq. 8) computes more comparisons of hypotheses from the same source. Corrections with no edits are filtered out so that the model can focus more on predicting the labels on edit words. Perfect corrections are also filtered out to maintain the label distribution balance. W&I+LOCNESS has 34,308 sentences and the resulting training data, obtained after filtering the unique corrections of the nine GEC systems above, has 65,824 hypotheses. During training, word labels and gap labels associated

with tokens not present in the source sentence are given a higher weight $z$ than other word and gap labels in the calculation of $\mathcal{L}_w$ and $\mathcal{L}_g$. We choose the hyper-parameters based on the model's performance on the BEA-2019 development set (Bryant et al., 2019) and the CoNLL-2013 test set (Ng et al., 2013). We list the hyper-parameters and explain our hyper-parameter search in Appendix B.

### 5.2 Evaluation

We evaluate our model on quality estimation, re-ranking, and system combination tasks. We compare our models to other GEC quality estimation models (NeuQE, VERNet, SOME) and a language model baseline, GPT-2 Large (Radford et al., 2019), which has been reported to perform relatively well on unsupervised GEC task (Alikaniotis and Raheja, 2019). We use the RC variant of NeuQE and the ELECTRA variant of VERNet which produce the highest scores on the CoNLL-2014 test set (Ng et al., 2014). For the system combination task, we compare our model to state-of-the-art GEC system combination methods in Section 2.2.

For the quality estimation and re-ranking tasks, we perform experiments in two scenarios: single system and multi-system evaluation. For single system evaluation, we follow Liu et al. (2021) on evaluating the models on the top-5 outputs of Riken&Tohoku (Kiyono et al., 2019) on the CoNLL-2014 test set. For multi-system evaluation, we evaluate the models on the outputs of 12 participating teams of the CoNLL-2014 shared task.

For the quality estimation task, we follow Chollampatt and Ng (2018b) in comparing the correlation coefficient of the quality score to the sentence-level $F_{0.5}$ score. We use Spearman's rank correlation coefficient (Spearman, 1904), which is the primary metric of the WMT-2022 shared task on quality estimation (Zerva et al., 2022).

For the re-ranking task, we pick one hypothesis with the highest quality score for each source

| Model | BEA-2019 Dev ($F_{0.5}$) | BEA-2019 Test | | | CoNLL-2014 Test | | |
|---|---|---|---|---|---|---|---|
| | | P | R | $F_{0.5}$ | P | R | $F_{0.5}$ |
| 1. T5-Large | 56.21 | 74.30 | 66.75 | 72.66 | 69.66 | 51.50 | 65.07 |
| 2. GECToR XLNet | 55.62 | 79.20 | 53.90 | 72.40 | 77.49 | 40.15 | 65.34 |
| 3. GECToR RoBERTa | 54.18 | 77.20 | 55.10 | 71.50 | 73.91 | 41.66 | 64.00 |
| 4. Riken&Tohoku | 53.95 | 74.7 | 56.7 | 70.2 | 73.26 | 44.17 | 64.74 |
| 5. UEDIN-MS | 53.00 | 72.28 | 60.12 | 69.47 | 75.15 | 41.21 | 64.52 |
| 6. Kakao&Brain | 53.27 | 75.19 | 51.91 | 69.00 | - | - | - |
| NeuQE | 29.30 | 68.48 | 20.19 | 46.32 | 66.48 | 15.87 | 40.59 |
| VERNet | 54.80 | 73.19 | 58.42 | 69.67 | 74.08 | 39.12 | 62.85 |
| SOME | 52.23 | 66.40 | 67.83 | 66.68 | 68.39 | 54.23 | 65.00 |
| GPT-2 | 52.00 | 67.20 | 68.08 | 67.38 | 68.30 | 52.65 | 64.47 |
| ESC | 63.09 | 86.65 | 60.91 | 79.90 | 81.48 | 43.78 | 69.51 |
| MEMT | 60.72 | 82.20 | 63.00 | 77.48 | 76.44 | 48.06 | 68.37 |
| EditScorer | 61.66 | 88.05 | 58.71 | 80.05 | 74.32 | 51.44 | 68.25 |
| GRECO | 60.74 | 80.03 | 66.22 | 76.83 | 76.39 | 50.35 | 69.23 |
| GRECO$_{voting}$ | 62.22 | 82.86 | 65.10 | 78.58 | 79.36 | 48.69 | **70.48** |
| GRECO$_{voting+ESC}$ | **63.40** | 86.45 | 63.13 | **80.50** | 79.36 | 48.69 | **70.48** |

Table 3: System combination results. The first group of rows shows the base GEC systems that are combined, while the second and the third group of rows show the combination results of existing quality estimation models and system combination methods respectively. We do not show Kakao&Brain's score on CoNLL-2014 as it is not used in the CoNLL-2014 combination. ESC and MEMT results are taken from (Qorib et al., 2022). GRECO$_{voting}$ refers to our model with voting bias, and GRECO$_{voting+ESC}$ refers to our model with voting bias and edit scores from ESC.

sentence, and then compute the corpus-level $F_{0.5}$ score for the chosen hypotheses of all source sentences. We also add the source sentence as one of the hypotheses so that a model has the option to not make any corrections if the hypotheses are bad. We use the M$^2$ Scorer (Dahlmeier and Ng, 2012) to compute the $F_{0.5}$ score.

For the system combination task, we combine the base systems given in Table 3 using ESC, MEMT, and EditScorer. We also combine the same base systems using beam search via quality estimation for the different quality estimation methods NeuQE, VERNet, SOME, GPT-2, GRECO, GRECO$_{voting}$, and GRECO$_{voting+ESC}$. We report the $F_{0.5}$ scores on the BEA-2019 test set and CoNLL-2014 test set.

We use William's test (Williams, 1959) to measure the statistical significance of the correlation coefficients. We use bootstrap resampling on 100 samples for the statistical significance of the $F_{0.5}$ scores in the re-ranking and system combination tasks.

## 6 Results

### 6.1 Quality Estimation and Re-Ranking

We report the results of quality estimation and re-ranking evaluation in Table 2. Our model sig-

nificantly outperforms all other quality estimation models on the correlation score and $F_{0.5}$ score in both experimental settings ($p < 0.001$). Our re-ranking result has higher precision, recall, and $F_{0.5}$ score compared to the top-1 output of Riken&Tohoku, which has a precision, recall, and $F_{0.5}$ score of 68.59, 44.87, and 62.03 respectively[3].

### 6.2 System Combination

We report the results of the system combination experiments in Table 3. Existing GEC quality estimation models fail to produce better corrections. More surprisingly, all of them produce combination scores that are lower than the fourth best base system on the BEA-2019 experiment and the second best base system on the CoNLL-2014 experiment. Our model without any additional biases (GRECO) successfully produces better corrections with 4.17 points and 3.89 points higher than the best base system on the BEA-2019 (T5-Large) and CoNLL-2014 (GECToR XLNet) test sets respectively.

By adding the voting bias, our model outperforms MEMT on both datasets and ESC and Ed-

---

[3] The top-1 performance for Riken&Tohoku in this experiment is different from Table 3 because the data for this experiment was from reproduction by Liu et al. (2021), which does not include right-to-left re-ranking due to that component not being publicly available according to them.

| Model | BEA-2019 Test | | |
|---|---|---|---|
| | **P** | **R** | **F$_{0.5}$** |
| 1. T5-XL | 76.85 | 67.26 | 74.72 |
| 3. GECToR-Large | 80.70 | 53.39 | 73.21 |
| + Model [2], [4], [5], [6] from Table 3 | | | |
| ESC | 86.64 | 61.54 | 80.10 |
| GRECO | 80.46 | 66.59 | 77.24 |
| GRECO$_{voting}$ | 83.24 | 65.12 | 78.85 |
| GRECO$_{voting+ESC}$ | 86.66 | 63.72 | **80.84** |

Table 4: Combination of the same base systems in Table 3, but with model [1] replaced by T5-XL and model [3] replaced by GECToR-Large RoBERTa.

itScorer on the CoNLL-2014 test set. Our model when augmented with voting bias and edit scores from ESC outperforms ESC and EditScorer on the BEA-2019 test set by 0.6 and 0.45 points respectively. Note that EditScorer is trained with 70 times more data than GRECO. Using edit scores from ESC on the CoNLL-2014 test set does not change the result since the optimal edit weight ($\beta$) is zero. In other experiments, we found that combining with ESC can also improve the $F_{0.5}$ score on the CoNLL-2014 test set (Appendix Table 11). Our final model has significantly higher scores than all other methods ($p < 0.005$).

We also evaluate our model on the combination of stronger GEC models. We replace T5-Large by T5-XL and GECToR RoBERTa by GECToR-Large RoBERTa (Tarnavskyi et al., 2022) from the base systems and achieve the highest BEA-2019 test score, 80.84, reported to date. (Table 4).

# 7 Discussions

This section discusses important characteristics of our model. From this section onward, we sometimes refer to our model GRECO$_{voting}$ as G$_v$, and GRECO$_{voting+ESC}$ as G$_{v+E}$.

## 7.1 Model-Agnostic

GRECO is model-agnostic, which means we can add or change the base systems during inference. We run an experiment of adding the C4-200M GEC system (Stahlberg and Kumar, 2021) to the base systems of the CoNLL-2014 system combination while using the model weights and hyperparameters from Table 3. This experiment setting is not possible with ESC and MEMT which require the same set of base systems during training and testing. In this experiment, we also replace

| Model | CoNLL-2014 Test | | |
|---|---|---|---|
| | **P** | **R** | **F$_{0.5}$** |
| 1. T5-XL | 74.40 | 52.02 | 68.50 |
| 7. C4-200M | 75.47 | 49.06 | 68.13 |
| + Model [2] – [5] from Table 3 | | | |
| GRECO | 76.68 | 51.54 | 69.86 |
| GRECO$_{voting}$ | 79.60 | 49.86 | **71.12** |

Table 5: Combination of the same base systems in Table 3, but with T5-Large replaced with T5-XL and the C4-200M model added to the base systems.

T5-Large with T5-XL. From this experiment, our model achieves the highest CoNLL-2014 test score reported to date, 71.12 (Table 5). We also evaluate it on the CoNLL-2014 test set with 10 annotations, using the same approach as Bryant and Ng (2015)[4] and report the highest score to date, 85.21.

## 7.2 Fluency

We analyze the output fluency of our methods by measuring the perplexity of the generated corrections using GPT-2, since prior work (Kann et al., 2018) has found that perplexity correlates with human fluency score. We found that our method generates more fluent corrections than all other methods, and adding more biases to our model makes the model less fluent (Figure 3). Based on the generated sentences, edit-based methods like ESC and EditScorer are too optimized toward picking the correct edits which can make the sentence unnatural. Our model with three modes of generality offers a flexible trade-off between $F_{0.5}$ score and fluency. Our GRECO$_{voting+ESC}$ achieves a higher $F_{0.5}$ score and better fluency at the same time compared to the previous best system combination methods ESC and EditScorer.

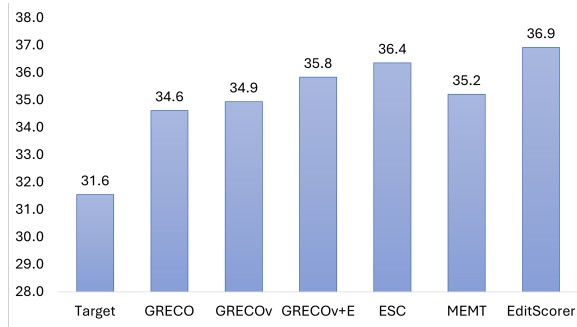

Figure 3: Median perplexity of the generated corrections on the BEA-2019 development set (lower is better).

[4]Evaluate the output on 10 sets of 9-annotation references and average the $F_{0.5}$ scores from all 10 sets.

| Model | $b$ | BEA-Dev | BEA-Test | CoNLL-2014 | | $b$ | BEA-Dev | BEA-Test | CoNLL-2014 |
|---|---|---|---|---|---|---|---|---|---|
| GRECO$_{voting+ESC}$ | 16 | 63.40 | 80.50 | 70.48 | | 1 | 62.96 | 80.18 | 67.48 |
| GRECO$_{voting+ESC}$ | 1 | 62.96 | 80.18 | 67.48 | | 2 | 63.13 | 80.17 | 69.38 |
| GRECO$_{voting}$ | 16 | 62.22 | 78.58 | 70.48 | | 4 | 63.31 | 80.31 | 70.25 |
| GRECO$_{voting}$ | 1 | 60.66 | 77.04 | 67.48 | | 8 | 63.34 | 80.47 | 70.36 |
| GRECO | 16 | 60.74 | 76.83 | 69.23 | | 12 | 63.34 | 80.49 | 70.34 |
| GRECO | 1 | 60.18 | 76.77 | 66.84 | | 16 | 63.40 | 80.50 | 70.48 |
| GRECO without rank loss | 16 | 59.78 | 75.62 | 68.51 | | 20 | 63.38 | 80.53 | 70.44 |
| GRECO without rank loss | 1 | 57.50 | 73.05 | 65.64 | | 24 | 63.33 | 80.53 | 70.36 |
| | | | | | | 32 | 63.37 | 80.55 | 70.56 |

Table 6: Ablation study for each component of the model (left) and different beam sizes $b$ (right).

## 7.3 Ablation

We run an ablation study to evaluate the contribution of each component of our method and the effect of beam size on model performance (Table 6). We found that both the training (rank loss) and inference (voting bias, edit scoring, beam search) techniques contribute to the final model performance. We also found that the performance does not change much with beam size >= 8. However, the difference between greedy (beam size = 1) and beam search (beam size = 16) decoding is quite substantial, especially on the CoNLL-2014 test set. Beam search has more effect on the CoNLL-2014 test set because its number of edits per sentence is more than the BEA-2019 development set.

## 7.4 Number of Base Systems

We investigate the performance of our model when the number of base systems is reduced. We use the base systems in Table 3 and randomly sample 5 combinations of base systems for each experiment (except when combining 5 systems where there is only one combination). We evaluate the combination on the CoNLL-2014 test set and calculate the average $F_{0.5}$ score for each number of base systems (Figure 4). We found that ESC's performance deteriorates rapidly when the number of base systems is reduced, while GRECO$_{voting}$ and EditScorer can maintain their performance.

## 8 Conclusion

In this paper, we present novel methods to utilize GEC quality estimation models for system combination with varying generality: model-agnostic, model-agnostic with voting bias, and model-dependent method. We report that existing GEC quality estimation models are not able to dif-

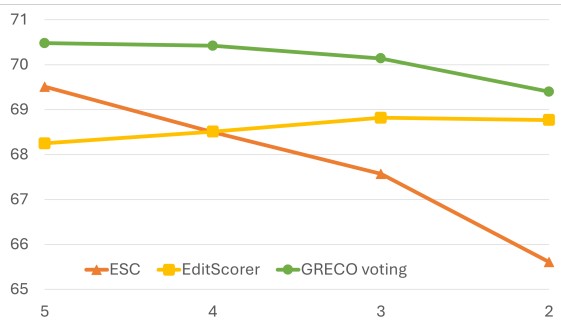

Figure 4: Average $F_{0.5}$ score on the CoNLL-2014 test set for varying number of base systems.

ferentiate good corrections from bad ones, which is shown by their ineffectiveness on the re-ranking and system combination tasks. Hence, there is a need for a new quality estimation model for GEC.

We present a new state-of-the-art quality estimation model, GRECO. Our model outperforms existing models on quality estimation and re-ranking evaluation. Our re-ranking of the top-5 hypotheses of Riken&Tohoku beats the performance of the top-1 hypothesis. Our source code and model weights are publicly available, making our model directly usable as a post-processing tool for re-ranking GEC systems' outputs.

Our model, combined with a voting bias and an edit-based system combination method, successfully improves the $F_{0.5}$ scores on the GEC system combination task and produces the highest $F_{0.5}$ scores on the CoNLL-2014 test set and BEA-2019 test set to date, which are 71.12 and 80.84 respectively.

## Limitations

Our model is trained on English GEC data with a single reference, and we only report experimental results on English GEC. Future work can apply our

model to other languages. We believe our work does not pose any risks to society.

## Acknowledgements

We thank the anonymous reviewers for their helpful comments. This research is supported by a research grant from TikTok (WBS No. A-8000972-00-00). The computational work for this article was partially performed on resources of the National Supercomputing Centre, Singapore (https://www.nscc.sg). Cloud resources involved in this research work were also partially supported by NUS IT's Cloud Credits for Research Programme and Amazon Web Services.

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

## A  Oracle Performance

In this section, we analyze the performance of an oracle system combination method on the BEA-2019 development set and CoNLL-2014 test set. Here, we assume the oracle always picks the correct edits from the union of all edits from the base systems, making the precision 100%. From Table 7, we see that the oracle combination of T5-Large and GECToR XLNet on the CoNLL-2014 test set already produces a very high score, 13.97 points higher

than our state-of-the-art $F_{0.5}$ score, while the combination of 5 systems is 16.56 points higher than our state-of-the-art $F_{0.5}$ score. This analysis shows that there is much room for further investigation on the GEC system combination task. Research on GEC system combination is important to better understand how and why GEC models have complementary strengths and why it is not trivial to combine them.

| Base systems | BEA Dev | CoNLL-2014 |
|---|---|---|
| T5-Large + GECToR-XLNet | 82.94 | 84.45 |
| +GECToR-RoBERTa | 84.35 | 85.05 |
| +Riken&Tohoku | 85.81 | 86.53 |
| +UEDIN-MS | 86.68 | 87.04 |

Table 7: The oracle $F_{0.5}$ scores with increasing number of base systems.

## B Hyper-Parameters

Our method introduces seven hyper-parameters (HP), one for training data generation ($n$), four for training the model ($\gamma$, $\mu$, dropout rate $d$ for the classifier layers, and weight $z$ for weighting the cross-entropy loss of word labels and gap labels associated with tokens not present in the source sentence), and two for inference ($\alpha$ and $\beta$). We also search the batch size ($bs$), learning rate ($lr$), and beam size ($b$). See Table 8. Note that $bs$ needs to be divisible by $n$. We search the hyper-parameters through manual tuning and select them based on the $F_{0.5}$ scores on the validation sets. For the inference hyper-parameters, we evaluate the model 9 times to find the optimal $\alpha$ on the validation sets, then another 9 evaluations (with the $\alpha$ applied) to find the optimal $\beta$.

### B.1 Training Data

As mentioned in Section 5.1, we arrange the training data in small groups of size $n$, in which hypotheses from the same source are put together. In our experiment, $n = 4$, so the number of rank loss computations within one group is $\binom{n}{2} = \binom{4}{2} = 6$.

### B.2 Final Hyper-Parameters

We use DeBERTa-V3-Large[5] as the backbone of our model, which has about 434M parameters. Our model has a word classifier with 1M parameters

[5] https://github.com/microsoft/DeBERTa

| HP | Bound | Search Interval |
|---|---|---|
| $lr$ | $lr < 1$ | $\{1, 2, 3\} \times 10^{-5}$ |
| $bs$ | $bs > 1$ | 32, 64, 128, 256 |
| $n$ | $1 \leq n \leq bs$ | 4, 8 |
| $\gamma$ | $0 \leq \gamma \leq 1$ | 0, 0.2, 0.3, 0.5, 1.0 |
| $d$ | $0 \leq d < 1$ | 0.2, 0.25, 0.3 |
| $z$ | $z \geq 0$ | 2.0 |
| $\mu$ | $\mu > 0$ | 1, 4, 5, 7 |
| $\alpha$ | $0 \leq \alpha \leq 1$ | 0.1, 0.2, ..., 0.9 |
| $\beta$ | $0 \leq \beta < 1$ | 0.1, 0.2, ..., 0.9 |
| $b$ | $b \geq 1$ | 1, 2, 4, 8, 12, 16, 20, 24, 32 |

Table 8: Hyper-parameters introduced by our method.

and a gap classifier with 1M parameters. In total, our model has 436,113,410 effective parameters. We detail the hyper-parameters of our final model in Table 9.

| HP | Value |
|---|---|
| $lr$ | $2 \times 10^{-5}$ |
| $lr$ scheduler | linear |
| $bs$ | 128 ($32 \times 4$ gradient acc) |
| Optimizer | AdamW ($\beta_1 = 0.9$, $\beta_2 = 0.999$) (Loshchilov and Hutter, 2019) |
| Max epochs | 15 |
| $n$ | 4 |
| $\gamma$ | 0.2 |
| $d$ | 0.25 |
| $z$ | 2.0 |
| $\mu$ | 5 |
| $\alpha$ | 0.4 |
| $\beta$ | 0.7 for BEA-2019 in Table 3 0.0 for CoNLL-2014 in Table 3 0.6 for BEA-2019 in Table 4 |
| $b$ | 16 |

Table 9: Hyper-parameters of our final model. Our model converged after epoch 4.

## C Running Time

We run all of our experiments on a single NVIDIA A100 40GB GPU. We report the training and testing time in Table 10.

## D Experimental Results

We repeat our experiments five times with different random seeds for training the models. We report the $F_{0.5}$ scores in Table 11 and report the mean and standard deviation in Table 12.

| Step | Running Time (HH:MM:SS) |
|------|------------------------|
| Training | 09:53:52 |
| Inference | |
|   BEA-2019 Dev | 00:06:08 |
|   BEA-2019 Test | 00:05:22 |
|   CoNLL-2013 | 00:01:46 |
|   CoNLL-2014 | 00:01:51 |

Table 10: Running time of each process on a single NVIDIA A100 40GB GPU.

| No | BEA-2019 Test | | |
|----|--------|-----------|--------------|
| | GRECO | $GRECO_v$ | $GRECO_{v+E}$ |
| 1 | 76.83 | 78.58 | 80.50 |
| 2 | 76.74 | 78.54 | 80.42 |
| 3 | 76.00 | 77.94 | 80.17 |
| 4 | 76.31 | 78.43 | 80.63 |
| 5 | 76.78 | 78.93 | 80.49 |
| No | CoNLL-2014 Test | | |
| | GRECO | $GRECO_v$ | $GRECO_{v+E}$ |
| 1 | 69.23 | 70.48 | 70.48 |
| 2 | 68.81 | 70.31 | 70.24 |
| 3 | 68.48 | 69.60 | 70.08 |
| 4 | 68.38 | 69.77 | 69.77 |
| 5 | 68.57 | 69.83 | 70.45 |

Table 11: The details of our experimental results. The cell values are the $F_{0.5}$ scores. Run 1 has the best score on the development set so it was chosen in our experiments.

| Model | BEA-Test | CoNLL-14 |
|-------|----------|----------|
| ESC | 79.86±0.07 | 69.47±0.14 |
| MEMT | 76.66±0.82 | 68.14±0.20 |
| $GRECO_{voting+ESC}$ | 80.44±0.17 | 70.20±0.29 |

Table 12: Average and standard deviation of $F_{0.5}scores(\bar{x} \pm \sigma)$ on the BEA-2019 test set and CoNLL-2014 test set from 5 experiments. The ESC and MEMT results are taken from (Qorib et al., 2022). Sorokin (2022) only reported a single data point, so we do not include EditScorer in this table.

# E Resources

We list the data sources in Table 13. We use the standard train/validation/test splits for all datasets. We list the GEC quality estimation model sources in Table 14, the GEC model sources in Table 15, and the scorer source code sources in Table 16.

| Data | URLs |
|---|---|
| 1. W&I+LOCNESS | https://www.cl.cam.ac.uk/research/nl/bea2019st/ |
| 2. BEA-2019 Dev | https://www.cl.cam.ac.uk/research/nl/bea2019st/ |
| 3. BEA-2019 Test | https://www.cl.cam.ac.uk/research/nl/bea2019st/ |
| 4. CoNLL-2013 Test | https://www.comp.nus.edu.sg/~nlp/conll13st.html |
| 5. CoNLL-2014 Test | https://www.comp.nus.edu.sg/~nlp/conll14st.html |
| 6. Participants of CoNLL-2014 | https://www.comp.nus.edu.sg/~nlp/conll14st.html |
| 7. Top-5 outputs of Riken&Tohoku | https://github.com/thunlp/VERNet/tree/main |

Table 13: Data sources.

| Model | URLs |
|---|---|
| 1. NeuQE | https://github.com/nusnlp/neuqe |
| 2. VERNet | https://github.com/thunlp/VERNet/ |
| 3. SOME | https://github.com/kokeman/SOME |
| 4. GPT-2 | https://github.com/openai/gpt-2 |

Table 14: Quality estimation models.

| Model | URLs |
|---|---|
| 1. T5-XL | https://github.com/google-research-datasets/clang8 |
| 2. T5-Large | https://github.com/google-research-datasets/clang8 |
| 3. GECToR-Large | https://github.com/MaksTarnavskyi/gector-large |
| 4. GECToR XLNet | https://github.com/grammarly/gector/tree/fea1532608 |
| 5. GECToR RoBERTa | https://github.com/grammarly/gector/tree/fea1532608 |
| 6. Riken&Tohoku | https://github.com/butsugiri/gec-pseudodata |
| 7. UEDIN-MS | https://github.com/grammatical/pretraining-bea2019/ |
| 8. Kakao&Brain | https://github.com/kakaobrain/helo_word/ |
| 9. BART-GEC | https://github.com/Katsumata420/generic-pretrained-GEC |

Table 15: GEC systems' sources.

| Scorer | URLs |
|---|---|
| 1. ERRANT | https://github.com/chrisjbryant/errant |
| 2. M2Scorer | https://www.comp.nus.edu.sg/~nlp/conll14st.html |

Table 16: Scorer source code sources.