# OpenReview forum: "System Combination via Quality Estimation for Grammatical Error Correction"
_EMNLP/2023/Conference — EMNLP 2023 Main_

### Official Review · Reviewer_8ah8 · 2023-08-03

**Soundness:** 4

**Excitement:**

4: Strong: This paper deepens the understanding of some phenomenon or lowers the barriers to an existing research direction.

**Paper Topic And Main Contributions:**

This paper presented a system combination method, which selects the best combination of edits output from multiple grammatical error correction (GEC) systems.

Basically, the method computed quality scores when the edits are applied to the source sentence and searched for the best combination using beam search.  The quality score, which is based on word scores of substitution and deletion and gap scores of insertion, was computed by the pretrained encoder that learned the order of F_{0.5} scores.  The proposed system also used the voting score and the scores output from element GEC systems.

In their experiments, F_{0.5} score of the proposed method was better than those of each QE system. In comparison with other system combination methods, the proposed method achieved the highest scores on the BEA-2019 and CoNLL-2014 test sets.

**Questions For The Authors:**

A) Sec. 3.1 and 3.2:
The proposed method tested applicability of each edit one-by-one.  If there are simultaneous edits such as agreement, is there any concern that only one of them will be applied?  Was the beam size 16 enough?

B) Lines 274-284:
It looks the unit size of the intermediate layer was not written anywhere.  What were the sizes of $A_{w}$, $A_{g}$, $a_{w}$, and
$a_{g}$?

C) Eq. 8:
The combination number of $y^{r}_{v} > y^{r}_{w}$ becomes huge if the
number of element systems increases.  Were the systems sampled to
reduce the computational cost?

D) Line 411:
Is this the same as 65,824 hypotheses?

E) Lines 476-477:
I could not find these values in Table 3.
If they are not shown in the table, please note it.

**Reasons To Accept:**

* Although the proposed method is a combination method of multiple GEC systems, it is model-agnostic, and therefore scalability and usability must be high.
* F_(0.5) score of the existing test set was the highest, and the experiments were carefully carried out.

**Reasons To Reject:**

I could not find crucial problems except for the questions written in the next section.

**Reproducibility:**

4: Could mostly reproduce the results, but there may be some variation because of sample variance or minor variations in their interpretation of the protocol or method.

**Reviewer Confidence:**

3: Pretty sure, but there's a chance I missed something. Although I have a good feel for this area in general, I did not carefully check the paper's details, e.g., the math, experimental design, or novelty.

**Typos Grammar Style And Presentation Improvements:**

* Because the proposed method is system combination, the result will be changed along with the combination of element systems. It is better to discuss the other combination of the element systems.

* It is interesting whether only the gold edits will be selected or not if the proposed system is applied to the mixture of the gold edits and edits output from multiple GEC systems.

---

> ### Author Rebuttal · Authors · 2023-08-29
>
> Thank you very much for your thorough review and for appreciating our work.
>
> Questions for the Authors:
>
> A. Like any application of beam search, there is always a risk of missing long dependencies. However, if the model can identify the correct edit well, there will be no issue, even with a very small beam size. We ran an ablation study for the beam search and found that the performance does not change much with beam size >= 8. We found that beam size = 16 is sufficiently good, and our model has a higher F0.5 score on the subject-verb agreement error type compared to ESC, EditScorer, and MEMT.
>
> Beam      | BEA-Dev | BEA-Test | CoNLL-Test
> ------------ | ------------- | -------------- | ---------
> Beam 1  | 62.96 | 80.18 | 67.48
> Beam 2  | 63.13 | 80.17 | 69.38
> Beam 4  | 63.31 | 80.31 | 70.25
> Beam 8  | 63.34 | 80.47 | 70.36
> Beam 12 | 63.34 | 80.49 | 70.34
> Beam 16 | 63.40 | 80.50 | 70.48
> Beam 20 | 63.38 | 80.53 | 70.44
> Beam 24 | 63.33 | 80.53 | 70.36
> Beam 32 | 63.37 | 80.55 | 70.56
>
> B. The size of $A_w$ and $A_g$ is model_dim x model_dim and the size of $a_w$ and $a_g$ is model_dim x 1. In our case, we use DeBERTaV3-Large which has a model_dim of 1024. We will clarify this in the paper.
>
> C. We group the hypotheses into smaller groups of size $n$ in Section 5.1 (second paragraph). In our experiment, we use $n=4$ (a hyper-parameter). Note that n is independent of the number of base (element) systems that we combine. This way, no matter how many systems we combine, the combination number of $y^{r}{v} > y^{r}{w}$ is capped to $n \choose 2$.
>
> D. That is correct.
>
> E. The scores of Riken&Tohoku in Table 3 are from the official model outputs of the Riken&Tohoku paper (Kiyono et al., 2019 [1]), where the model is an ensemble with right-to-left reranking. The values in line 477 (and the data used for the single-system reranking evaluation in Table 2) are from a reproduction without right-to-left reranking by Liu et al., (2021) [2]. Liu et al. mentioned that the right-to-left reranking model is not publicly available and is trained with unpublished data.
>
> Typos Grammar Style and Presentation Improvements:
>
> 1. We ran system combination experiments on the CoNLL-2014 test set with different numbers of base systems. We used the base systems in Table 3 and randomly sampled 5 combinations of base systems for each experiment (except when combining 5 systems where there was only one combination). We report the average F0.5 score for each number of base systems in the table below. We found that ESC’s performance deteriorates rapidly when the number of base systems is reduced, while HyRank+voting and EditScorer can maintain their performance.
>
> | # systems | ESC  | EditScorer  | Hyrank+voting |
> | ------------- | -------- | --------------- | ------------------- |
> | 5               | 69.51 | 68.25         | 70.48 |
> | 4               | 68.50 | 68.51         | 70.42 |
> | 3               | 67.57 | 68.82         | 70.14 |
> | 2               | 65.61 | 68.77         | 69.40 |
>
> 2. When we add gold edits to the base systems, we found that on the BEA-2019 development set, 77% of the selected edits are from the gold edits, while on the CoNLL-2014 test set, 80.69% of the selected edits are the gold edits.
>
> References:
>
> [1] Shun Kiyono, Jun Suzuki, Masato Mita, Tomoya Mizumoto, and Kentaro Inui. 2019. An Empirical Study of Incorporating Pseudo Data into Grammatical Error Correction. EMNLP.
>
> [2] Zhenghao Liu, Xiaoyuan Yi, Maosong Sun, Liner Yang, and Tat-Seng Chua. 2021. Neural Quality Estimation with Multiple Hypotheses for Grammatical Error Correction. NAACL.

---

### Official Review · Reviewer_nJd5 · 2023-08-05

**Soundness:** 3

**Excitement:**

3: Ambivalent: It has merits (e.g., it reports state-of-the-art results, the idea is nice), but there are key weaknesses (e.g., it describes incremental work), and it can significantly benefit from another round of revision. However, I won't object to accepting it if my co-reviewers champion it.

**Paper Topic And Main Contributions:**

The paper proposes a new GEC quality estimation model and three methods for utilizing GEC quality estimation models for system combination with varying generality: model agnostic, model-agnostic with voting bias, and model-dependent methods. The results suggest that the proposed method improves the F0.5 scores on the GEC system combination task and achieves the highest F0.5 scores on the CoNLL-2014 test set and the BEA-2019 test set.

**Reasons To Accept:**

1. The paper highlights the low performance of existing GEC quality estimation models when used for system combination and introduces a new method for system combination.
2. The paper achieves a new state-of-the-art F0.5 score on the BEA-2019 test set and CoNLL-2014 test set.
3. The proposed method for GEC system combination is model-agnostic, making it easy to apply to GEC systems.

**Reasons To Reject:**

1. Some details are not very clear:
- In the second paragraph of the introduction, the author uses the word "Thus," but the reasoning relationship is not evident.
- Figure 2 could be improved for better aesthetics.
- In the method section, the term "gap labels" should be clarified from the beginning, as it might cause confusion for readers.
- In section 6.1, it is not clear on which test set the comparison of reranking results and the output of Riken&Tohoku is based.

2. It will be more convincing to conduct a human evaluation experiment on the system combination results.

3. Reporting the time taken for system combination will be beneficial. Is the proposed method faster than previous work?

**Reproducibility:**

4: Could mostly reproduce the results, but there may be some variation because of sample variance or minor variations in their interpretation of the protocol or method.

**Reviewer Confidence:**

3: Pretty sure, but there's a chance I missed something. Although I have a good feel for this area in general, I did not carefully check the paper's details, e.g., the math, experimental design, or novelty.

---

> ### Author Rebuttal · Authors · 2023-08-29
>
> Thank you very much for your thorough review and for appreciating our work.
>
> 1.a. In this paragraph, what we meant is that since a GEC model that has a high corpus-level F0.5 score may occasionally still make mistakes, we need a quality estimation model to evaluate the corrections proposed by a GEC model before trusting its corrections in a real-world setting where we don’t have the gold reference.
>
> 1.b. and 1.c. Thank you for the suggestion. We will improve Figure 2 and the clarify gap labels at the beginning of Section 4.
>
> 1.d. We rerank the output of Riken&Tohoku (Kiyono et al., 2019) on the CoNLL-2014 test set as explained in Section 5.2 line 454. We will make this clearer by explaining it in Table 2’s caption.
>
> 2. The F0.5 scores from the M2-Scorer and ERRANT have been reported to highly correlate with human judgment [1,2], so we believe that reporting the F0.5 score is sufficient to demonstrate the models’ capability in combining GEC corrections.
>
> 3. We report our training and inference time in Appendix Table 14. The time needed for combining GEC systems using our model with beam=16 is fast, about 0.07s per sentence, even though it is not as fast as statistical methods like MEMT and ESC.
>
> Inference time (per sentence)
> * HyRank: 0.07 s
> * EditScorer: 0.06 s
> * ESC: 0.004 s
> * MEMT: 0.005 s
>
> References:
>
> [1] Peiyuan Gong, Xuebo Liu, Heyan Huang, and Min Zhang. 2022. Revisiting Grammatical Error Correction Evaluation and Beyond. EMNLP.
>
> [2] Roman Grundkiewicz, Marcin Junczys-Dowmunt, and Edward Gillian. 2015. Human Evaluation of Grammatical Error Correction Systems. EMNLP.

---

### Official Review · Reviewer_9qEn · 2023-08-05

**Soundness:** 5

**Excitement:**

4: Strong: This paper deepens the understanding of some phenomenon or lowers the barriers to an existing research direction.

**Paper Topic And Main Contributions:**

The paper proposes a new method of system combination for grammatical error correction. Given the edits of several basic models, the proposed model is trained to answer, whether the proposed edit is correct. It allows to improve significantly the quality of the basic model, also surpassing the previous methods of model combinations and even the current SOTA on English GEC. Besides this main idea, the work applies several clever solutions to model training and inference. The paper also contains deep analysis of different methods' properties.

**Questions For The Authors:**

A) in formula (1) in the Subsection 3.1 you say that the score of a hypothesis extended with incorrect edit should be lower than of the hypothesis itself. I think this is incorrect, consider the sentence "The game I plays today was interesting before I began to lose". The score of the edit plays->play is lower than the one of the plays->play, but probably higher than the score of the original hypothesis without edits.
B) you introduce several nonstandard components to the model, it would be useful to understand their proper role via ablation. First, I mean beam search for edits application. For example, (Sorokin, 2022) applies edits in a greedy fashion beginning from the most probable. I would be glad to know what part of your improvement over him is due to better training setup and what part belongs to improved inference.
C) the same ablation is required for ranking loss
D) probably, the model of (Sorokin, 2022) performs poorly on spelling data because it was trained on GECTOR edits which rarely include spelling by design of GECTOR. Can you provide statistics analogous to Table 10 for some other type of error?
E) probably, the model of (Sorokin, 2022) is rather not a method of model combination since it works with one model as well. May be, you should mention it.

**Reasons To Accept:**

* new SOTA on English GEC
* a new method of model combination by edits rescoring
* several other innovations, such as multicomponent loss for model training and beam-like inference are proposed
* thorough analysis of different models in terms of their quality and fluency

**Reasons To Reject:**

No reasons

**Reproducibility:**

4: Could mostly reproduce the results, but there may be some variation because of sample variance or minor variations in their interpretation of the protocol or method.

**Reviewer Confidence:**

5: Positive that my evaluation is correct. I read the paper very carefully and I am very familiar with related work.

**Typos Grammar Style And Presentation Improvements:**

A) if I understood correctly the subscript $Q_w$ in equation 8, here *w* refers to an incorrect hypothesis. But two equations higher *w* stands for "word". I advice to change the subscript to $Q_u$ or some other letter not used above in a different sense.
B) it is not easy to understand what dataset is used in Table 2, please mention it explicitly.

---

> ### Author Rebuttal · Authors · 2023-08-29
>
> Thank you very much for your thorough review and for appreciating our work.
>
> Questions for the Authors:
>
> A. It seems there is a typo in the example given in your review? Both edits in the example are the same. In our problem formulation in Section 3.1, we follow the philosophy of GEC that not producing any correction is better than producing a wrong correction; a hypothesis with a wrong edit, e.g., (Q(s, s⊕(plays -> play)), will have a lower F0.5 score than keeping the source sentence as is (Q(s, s)), since the wrong edit reduces the precision without increasing the recall. We believe that the quality estimation score should have the same property as the official GEC score.
>
> B. and C. We have run an ablation study for the beam search. We found that the performance does not change much with beam size >= 8. However, the difference between greedy (beam 1) and beam 16 is quite substantial, especially on the CoNLL-2014 test set. Beam search has more effect on the CoNLL-2014 test set since the number of edits per sentence is more than the BEA-2019 dev set. The rank loss also has a considerable effect on the model performance.
>
> | Model                                       | Beam | BEA-Dev | BEA-Test | CoNLL-2014 test set |
> | ----------------------------------- | ------- | ----------- | ----------- | --------------- |
> | With rank loss + voting + ESC | 16       | 63.40 | 80.50 | 70.48 |
> | With rank loss + voting + ESC | 1         | 62.96 | 80.18 | 67.48 |
> | With rank loss + voting            | 16       | 62.22 | 78.58 | 70.48 |
> | With rank loss + voting            | 1         | 60.66 | 77.04 | 67.48 |
> | With rank loss                          | 16       | 60.74 | 76.83 | 69.23 |
> | With rank loss                          | 1         | 60.18 | 76.77 | 66.84 |
> | Without rank loss                     | 16       | 59.78 | 75.62 | 68.51 |
> | Without rank loss                     | 1         | 57.50 | 73.05 | 65.64 |
>
> We believe both our training and inference setups contribute to the performance gain. We will add this analysis to our paper.
>
> D. We agree that it may be the cause. We also found that the recall of our model on the R:VERB:SVA (edits that fix the verb form to conform to subject-verb agreement) error type performs much better than ESC and EditScorer, and is comparable to MEMT. See the table below. Both our model and MEMT have a similar beam search method that takes into account the interaction of edits. Even though MEMT has a slightly higher recall than our HyRank+voting+ESC method, its precision (70.11) is much lower than our model (77.07), making the F0.5 score of our model on this error type (77.96) much higher than MEMT (72.27).
>
> HyRank | Hyrank+voting | Hyrank+voting+ESC | ESC  | MEMT | EditScorer
> ----------- | ------------------- | --------------------------- | ------- | -------- | ---------------
> 77.7       | 79.73               | 81.76                        | 78.38 | 82.43  | 66.22
>
> E. Yes, we will mention this point.
>
> Typos Grammar Style and Presentation Improvements:
>
> A. Thank you for your suggestion on Equation 8. We will follow your suggestion.
>
> B. We use the CoNLL-2014 test set to evaluate the re-ranking performance in Table 2, as mentioned in lines 452 – 457. We are sorry if this was not clear. We will also add this explanation to Table 2’s caption directly to improve clarity.

---

### Meta-Review · Area_Chair_c1V6 · 2023-09-23

**Recommendation:** 5

**Metareview:**

The paper proposes a novel quality estimation model for GEC that has a higher correlation with the F0.5 score used to evaluate GEC model quality. Then they propose 3 methods of system combination based on edit rescoring and beam search, that use the QE model, and show new SOTA results on two English benchmarks.

The paper is well-written, the experimental results are solid, with new SOTA on English benchmarks. The paper also offers a thorough analysis of model outputs. It would be interesting to see how this work applied for other languages.

---

### Decision · Program_Chairs · 2023-10-07

**Decision:**

Accept-Main

**Comment:**

The paper proposes a novel quality estimation model for GEC that has a higher correlation with the F0.5 score used to evaluate GEC model quality. Then they propose 3 methods of system combination based on edit rescoring and beam search, that use the QE model, and show new SOTA results on two English benchmarks.

The paper is well-written, the experimental results are solid, with new SOTA on English benchmarks. The paper also offers a thorough analysis of model outputs. It would be interesting to see how this work applied for other languages.